# Time Variant Multi-Objective Interval-Valued Transportation Problem in Sustainable Development

**Gurupada Maity** [1,†], **Sankar Kumar Roy** [1,*] **and Jose Luis Verdegay** [2,†]

1    Department of Applied Mathematics with Oceanology and Computer Programming, Vidyasagar University, Midnapore 721102, West Bengal, India; maity.g0795@gmail.com
2    Department of Computer Science and Artificial Intelligence, University of Granada, 18071 Granada, Spain; verdegay@decsai.ugr.es
*    Correspondence: sankroy2006@gmail.com
†    These authors contributed equally to this work.

**Abstract:** Sustainable development is treated as the achievement of continued economic development without detriment to environmental and natural resources. Now-a-days, in a competitive market scenario, most of us are willing to pay less and to gain more in quickly without considering negative externalities for the environment and quality of life for future generations. Recalling this fact, this paper explores the study of time variant multi-objective transportation problem (MOTP) with consideration of minimizing pollution. Time of transportation is of utmost importance in reality; based on this consideration, we formulate a MOTP, where we optimize transportation time as well as the cost function. The parameters of MOTP are interval-valued, so this form of MOTP is termed as a multi-objective interval transportation problem (MOITP). A procedure is taken into consideration for converting MOITP into deterministic form and then for solving it. Goal programming is applied to solve the converted transportation problem. A case study is conducted to justify the methodology by utilizing the environmental impact. At last, conclusions and future research directions are included regarding our study.

**Keywords:** transportation problem; sustainability; interval number; time variant parameter; goal programming

## 1. Introduction

Sustainability is not about threat analysis; it is a system analysis. Specifically, it is about how environmental, economic, and social systems interact to their mutual advantage or disadvantage at various space-based scales of operations. Transporting goods from one place to another is the natural human activity observed from ancient days until today. Present scenarios of transportation are totally under the umbrella of automation created by science. In transportation, several vehicles are used, which run over the head of distinct fuels for communicating energy. For that energy communication, there are several types of gases exerted by vehicles, creating pollution in the atmosphere. In most of these cases, the fact of pollution is not considered by the agency, person, or governing body; they just focus on maximizing profit or on minimizing expenditure. However, in most of situations, everyone forgets to keep the atmosphere pollution free for the upcoming future. Therefore, it is time to think about minimizing pollution and to optimize the other objectives as much as possible, considering our valuable time in the system.

The classical Transportation Problem (TP) can be described to a special case of the linear programming problem, and its model is applied to determine how many units of commodity of goods should be shipped from each origin to various destinations, satisfying supply and demand

constraints, while it optimizes total cost of transportation. In a broad sense, TP is used in economic development, industrial management, both passenger and freight transportation modes, etc., and they have a critical role for the economic development of numerical countries. Single-objective TP is not adequate to solve complicated scenarios in real-life situations; therefore, we consider MOTP (multi-objective optimization problem), which has been applied in many fields of science, including engineering, economics, and logistics where optimal decisions need to be taken in the presence of trade-offs between two or more conflicting objectives. Minimizing cost and maximizing comfort while buying a car and maximizing performance whilst minimizing fuel consumption and emission of pollutants of a vehicle are examples of multi-objective optimization problems involving two and three objectives, respectively. In most of real-life transportation systems, transportation cost is paid by the purchasers and selling the items gets benefits in favor of the supplier. Therefore, suppliers would like to optimize benefit and, at the same time, the purchasers would like to minimize cost of objective functions. Therefore, a conflicting situation occurs in the system. Furthermore, in a transportation system, if we consider production at the supplying origins, then there are some energies required to produce items; at the time of delivering the items, when vehicles are used, and during energy consumption or transportation, the environment is polluted. Hence, there is one more objective function added in the transportation, namely, to minimize pollution, which is required to consider a TP in multi-objective environment.

The transportation cost, the amount of goods available at the supply points, and the amounts required at the demand points are the parameters in the transportation problem. In earlier days, the transportation problem was developed with the assumption that the supply, demand, and cost parameters were exactly known. However, in real-life applications, all the parameters of the transportation problem are not generally defined precisely. It may have an interval value. Similar considerations may be taken for supply and demand parameters in TP of this paper. Keeping this point of view, this paper is designed with these parameters of the transportation problem as interval values.

For an MOTP, no single solution exists that simultaneously optimizes each objective function. In that case, the objective functions are said to be conflicting, and there exists a number of (possibly infinite) Pareto optimal solutions. A solution is called Pareto optimal if none of the objective functions can be improved in value without degrading some of the other objective values. There are several approaches to finding Pareto optimal solution of MOTP such as fuzzy programming, utility function approach, conic scalarization approach, weighting methods, etc. As a consequence, Pareto optimal solution cannot optimize expected achievement levels of a decision maker (DM). Goal programming minimizes the deviation between the achievement goals and selected achievement levels of DM, and here, this optimal solution is treated as a compromise solution. Goal Programming (GP), an analytical approach, is considered to solve the decision-making problem where targets have been assigned to all objective functions which are conflicting and noncommensurable to each other and DM interests to maximize the achievement level of the corresponding goals. Our mathematical model of multi-objective decision making can be considered in the following form:

$$\text{minimize} \quad \sum_{t=1}^{K} w_t |Z^t(x) - g_t|$$
$$\text{subject to} \quad x \in F,$$

where $F$ is the feasible set, $w_t$, $(t = 1, 2, \ldots, K)$ is the weight attached to the deviation of the achievement function, $Z^t(x)$ is the $t$th objective function to the $t$th goal, $g_t$ is the aspiration level to the $t$th goal, and $|Z^t(x) - g_t|$ represents the deviation of the $t$th goal. Let us take $d_t^+ = Z^t(x) - g_t$ if $Z^t(x) \geq g_t$; otherwise, $d_t^+ = 0$ $(t = 1, 2, \ldots, K)$. Also, we put $d_t^- = g_t - Z^t(x)$ if $Z^t(x) \leq g_t$; otherwise, $d_t^- = 0$. Then, $Z^t(x) - g_t = d_t^+ - d_t^-$, which implies that $Z^t(x) - d_t^+ + d_t^- = g_t$. Furthermore, $|Z^t(x) - g_t| = d_t^+ + d_t^-$. Thus, the model GP reduces to as follows:

$$\textbf{Model GP:} \quad \text{minimize} \quad \sum_{t=1}^{K} w_t(d_t^+ + d_t^-)$$

$$\text{subject to} \quad Z^t(x) - d_t^+ + d_t^- = g_t,$$

$$d_t^+ \geq 0, d_t^- \geq 0 \ (t = 1, 2, \ldots, K),$$

$$x \in F,$$

where $d_t^+$ and $d_t^-$ are over- and underachievements of the $t$th goal, respectively.

In most real-life situations, DM fails to put proper goals and, then using the experience of DM, consider goals by selecting an interval $[g_t^l, g_t^u]$. In this situation, consider $g_t = g_t^l$ for minimization problem and $g_t = g_t^u$ for maximization problem in the GP model. Another important factor is that, if values of $Z^t(x)$ are in different scales and there are big differences among the real numbers $g_t^u - g_t^l$, $\forall t$, then we reduce the weights $w_t$ by $w_t = w_t/(g_t^u - g_t^l)$ in the GP model. Based on the study of solving time variant MOTP with interval parameters, we consider three-fold aims:

(i)  A new way in which MOTP is solved by considering interval parameters with time.
(ii) The study of the proposed MOTP finds the optimal compromise solution along with the optimal time of transportation, although time is minimized in the transportation problem without considering any objective function corresponding to time for extracting the optimal solution.
(iii) The advantage of goal programming is utilized to solve MOITP.

The residue of this paper can be depicted as follows: An updated review on research in connection to this proposed study, the problem background, and the development of mathematical model of the study are presented in Sections 2–4, respectively. Section 5 contains the solution procedure along with two subsections. A reduction procedure from intervals to real numbers is presented in Section 5.1. An algorithm for solving the proposed MOITP is offered in Section 5.2. A case study to justify our present study is conducted in Section 6, which includes two subsections. Discussion on results and sensitivity analysis are carried out in Sections 6.1 and 6.2, respectively. Finally, concluding remarks and outlook of the study are described in Section 7.

## 2. Review of Related Works

The transportation model was first studied by Kantorovich [1], who had prescribed an incomplete algorithm for obtaining the solution of the TP. Hitchcock [2] considered the problem of minimizing cost of distribution of products from several factories to a number of customers. He developed a procedure to solve the TP, which closely resembles the simplex method for solving TP as the primal simplex transportation method developed by Dantzig [3]. Aneya and Nair [4] solved a bi-criteria TP in 1979. Optimization in passenger transportation and passenger flow control has been studied by Liu and Chen [5] and Wagenaar et al. [6], respectively. Zhen et al. [7] studied optimization in channel flow control. Xiang et al. [8] studied assignment problem, which is a special case of TP under uncertainty. The time-minimizing TP has been studied by several researchers: Hammer [9], Roy and Maity [10], Sharma and Swarup [11], Szware [12], and Bhatia et al. [13]. Liu [14] discussed a method for solving the cost minimization TP with varying demand and supply. A study on iterative algorithm for two-level hierarchical time minimization TP has been introduced by Sharma et al. [15]. Salazar et al. [16] proposed an algorithm for solving a bi-objective transportation location routing problem. Damci-Kurt et al. [17] solved a TP with market choice. A study on time-dependent fuzzy random location-scheduling transportation programming for hazardous materials has been done by Meiyi et al. [18]. In connection with minimizing pollution, a number of studies has been entitled by researchers such as Dulebenets [19], Das and Roy [20], Jalalian et al. [21], Dulebenets [22], and many others. Ebrahimnejad [23] proposed an improved approach for solving fuzzy TP with triangular fuzzy numbers. Prakash [24] solved a transportation problem with objectives to minimize total

cost and duration of transportation. An approach to solve a bi-level time-minimizing TP is studied by Sonia et al. [25]. Lei et al. [26] solved a problem on transportation cost allocation under the consideration of a fixed route. Vincent et al. [27] studied an interactive approach for the multi-objective TP with interval parameters. Roy et al. [28] studied MOTP under intuitionistic fuzzy environment. Roy and Midya [29] presented a study on multi-objective fixed-charge solid transportation problem under intuitionistic fuzzy environment.

Instead of a single choice, if there may be involved several choices associated with the transportation parameters like cost, supply, or demand, then DM confuses the selection of the proper choice for the parameters. In this circumstance, the study of TP creates a new direction which is called multi-choice transportation problem. Chang [30] proposed a multi-choice goal programming approach to solve mathematical programming. Again Chang [31] proposed another multi-choice goal programming approach in revised form, though the multi-choice concept discussed in both the papers of References [30,31] are totally related to the goals of objective functions. Maity et al. [32] presented a study for solving neural networking problem using TP. Mahapatra et al. [33] discussed a multi-choice stochastic transportation problem involving exponential distribution and extreme value distribution in which the multi-choice concept involved only in the cost parameters. Midya et al. [34] introduced a study on multi-item multi-objective transportation problem under uncertain environment. A study on goal programming for solving MOTP has been presented by Maity and Roy [35]. Maity et al. [36] studied MOTP under dual hesitant fuzzy environment. Roy et al. [37] studied conic scalarization for solving MOTP. Both ways multi-objective TP was solved by Roy et al. [38]. Maity et al. [39] introduced the cost reliability in MOTP and solved it using goal programming.

## 3. Problem Background

This study deals with the time minimization in TP under multi-objective environment. It is generated under the suitable consideration of sustainable development. Basically, TP is considered to find the optimal transportation cost. In the traditional model of TP [2], the only parameters are transportation cost, supply, and demand. The mathematical model of TP is defined as follows:
**Model 1**

$$\text{minimize} \quad z = \sum_{i=1}^{m} \sum_{j=1}^{n} C_{ij} x_{ij}$$

$$\text{subject to} \quad \sum_{j=1}^{n} x_{ij} \leq a_i \quad (i = 1, 2, \ldots, m),$$

$$\sum_{i=1}^{m} x_{ij} \geq b_j \quad (j = 1, 2, \ldots, n),$$

$$x_{ij} \geq 0 \ \forall \ i \text{ and } j,$$

where $C_{ij}$ $(i = 1, 2, \cdots, m; j = 1, 2, \cdots, n)$ represents the transportation cost per unit commodity from the $i$th origin to the $j$th destination and $x_{ij}$ $(i = 1, 2, \cdots, m; j = 1, 2, \cdots, n)$ is the decision variable which determines the amount of commodity transported from the $i$th origin to the $j$th destination. Here, $a_i$ $(i = 1, 2, \cdots, m)$ and $b_j$ $(j = 1, 2, \cdots, n)$ are availability and demand in $i$th origin and $j$th destination, respectively, and $\sum_{i=1}^{m} a_i \geq \sum_{j=1}^{n} b_j$ is the feasibility condition. A network of TP is shown in Figure 1.

Most of the researchers confined their minds by considering the concept of the referred parameters, but here, we are not restricted only with this concept. The objective function in a TP is usually considered as minimizing total transportation cost. In view of real-life situations, TP with a single-objective function is unable to tackle optimal decisions under the appearance of more than one objective function that conflict with each other. Based on this fact, we consider multiple objective functions in transportation situation such as minimizing total transportation cost, maximizing profit, and minimizing pollution factor along with the consideration of time which is to be minimized in

a single frame of optimization model. The parameters of each of the objective function are taken as interval valued.

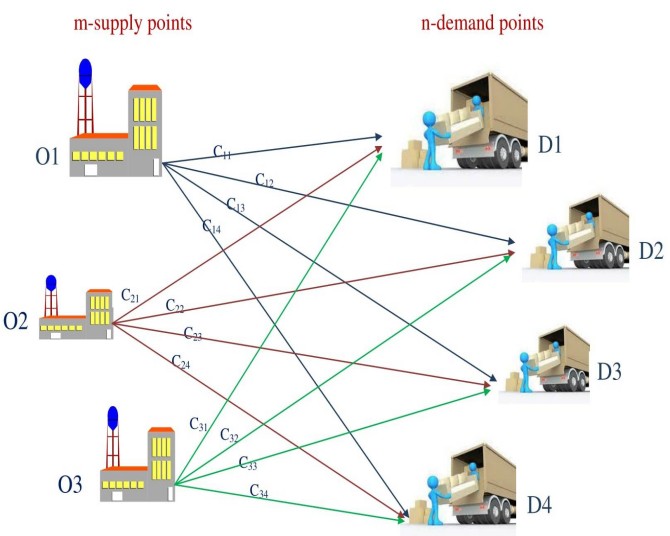

**Figure 1.** Graphical network of the transportation problem (TP).

Transportation of goods are highly related with time: if a customer wishes to get goods in a quicker time, he/she has to pay more transportation cost. Therefore, it is difficult to tackle the situation under the crisp transportation cost, and hence, we choose the transportation cost as an interval valued which depends on time.

During the transportation, vehicles frequently used run through power/fuel. In any transportation system, there are few poisonous gases exerted into the atmosphere directly or indirectly, so this pollution factor is considered as carbon emission. Therefore, the pollution factor is taken as one of the objective functions in this study, for minimizing the carbon emission. Again, for production system, energy is required, so the pollution factor such as carbon emission is directly or indirectly involved in the system. Again, DM wants to optimize the objective functions in their way: it may be the seller or customer. However, to save nature, it is utmost essential to make a decision minimizing the pollution factor of carbon emission. In this regard, we incorporate an objective function for minimizing pollution factor as carbon emission, of which the penalties are also taken as interval valued.

Furthermore, to optimize the multi-objective optimization problem, we introduce the goal programming approach. Considering sustainable development of nature, we are to consider minimization of the pollution factor in one of the important goals. In this study, time minimization is another objective function. However, the objective function regrading time is optimized without considering it like other objective functions.

## 4. Mathematical Model

Considering the traditional model of TP, here, we develop the mathematical model of MOITP. Due to some unavoidable market situations or for the cause of some special concessions to the customers in business ground, there may exist some cases that the transportation parameters in TP are not crisp values but may lie in an interval. Due to some unpredictable situations, such as weather condition, variation in share market, or unexpected demands in the market etc., both the purchaser and the supplier predict the amount of buying and selling goods, so it becomes an interval valued type.

Based on these phenomena, we consider the parameters of TP as interval valued type, and then the corresponding mathematical model (cf. Roy and Maity [10]) is defined as follows:

**Model 2**

$$\text{minimize} \qquad z = \sum_{i=1}^{m} \sum_{j=1}^{n} C'_{ij} x_{ij}$$

$$\text{subject to} \qquad \sum_{j=1}^{n} x_{ij} \leq a'_i \quad (i = 1, 2, \ldots, m), \qquad (1)$$

$$\sum_{i=1}^{m} x_{ij} \geq b'_j \quad (j = 1, 2, \ldots, n), \qquad (2)$$

$$x_{ij} \geq 0 \ \forall \ i \text{ and } j. \qquad (3)$$

Here, the parameters $C'_{ij}$, $a'_i$ and $b'_j$ are interval numbers, and they are defined as $C'_{ij} = [C^l_{ij}, C^u_{ij}]$, $a'_i = [a^l_i, a^u_i]$, and $b'_j = [b^l_j, b^u_j]$, where $C^l_{ij}$ and $C^u_{ij}$ are the lower bound and upper bound of the interval and for others.

Due to increasing complexities in the business era, the single objective function is no more a long lasting approach to tackle decision making situations in TP. Therefore, we incorporate the multiple objective functions in TP and formulate MOTP with interval cost parameter as follows:

**Model 3**

$$\text{minimize} \qquad z^k = \sum_{i=1}^{m} \sum_{j=1}^{n} C'^k_{ij} x_{ij}, \ \ k = 1, 2, \ldots, K$$

$$\text{subject to} \qquad \text{the constraints (1)–(3).}$$

In TP, time of transportation, especially for transporting the goods considering the sustainable development of nature, it is an important factor. In most of government and private industrial systems, the main aim for transporting goods is to reduce transportation cost and to optimize benefits within a quicker time. However, in many situations, no one would like to consider the optimal situation of controlling pollution during manufacturing as well as transportation. Therefore, in a faster world, to keep our nature in the best condition, it is the time to think not only of the benefit but also of the sustainability of nature during the system. As time is an important factor, we construct another objective function to minimize the transportation time as follows:

$$\text{minimize} \qquad T = \sum_{i=1}^{m} \sum_{j=1}^{n} T_{ij} \chi_{ij},$$

$$\text{where} \qquad \chi_{ij} = \begin{cases} 0, & \text{if } x_{ij} = 0 \ in \ X \\ 1, & \text{if } x_{ij} \neq 0 \ in \ X \end{cases} \qquad (4)$$

Here, $T_{ij}$ is the time of transporting the goods from the $i$th node to the $j$th destination and $X \in F'$; where $F'$, the set of all points satisfying constraints (1)–(3), is the feasible region of Model 3.

Hence, in our proposed model, we include multi-objective function along with objective function of time to be minimized and is defined as follows (see Model 4):

**Model 4**

$$\text{optimize} \qquad z^k = \sum_{i=1}^{m} \sum_{j=1}^{n} C'^k_{ij} x_{ij}, \ \ k = 1, 2, \ldots, K$$

$$\text{minimize} \qquad T = \sum_{i=1}^{m} \sum_{j=1}^{n} T_{ij} \chi_{ij},$$

$$\text{subject to} \qquad \text{the constraints } (1) - (3) \ \& \ (4).$$

Most of the researchers solved the multi-objective decision-making problem, and they obtained the compromise solution in a traditional way. In time-minimizing MOTP, optimization function of time is also taken as an objective function. However, here, we deal with time in an objective function as a new way. In the present study, time is correlated with the cost parameter of objective function $z^k$ in the following way: completion of production of goods in lesser time increases the cost of quantity; completion of transportation in lesser time increases the cost of transportation; and completion of the entire system in lesser time increases the pollution factor. Here, we incorporate a function to reduce the interval in such a way that the compromise solution of the objective function $z^k$ provides the solution of the time-minimizing objective function $T$. The procedure to find a compromised solution for the objective functions $z^k$ and $T$ by solving the objective function $z^k$ has been discussed in detail in the next section.

## 5. Solution Procedure

The mathematical model of MOTP described in this paper (see Model 3) cannot be solved directly due to presence of interval-valued parameters. Therefore, at first, we transfer the problem into MOTP with crisp penalties. After that, goal programming technique is imposed to determine the optimal compromise solution of the reduced MOTP.

Now, we define a function which depends on time of which the value lies in the interval $[0, 1]$, and this is used to reduce the interval valued cost parameter to the real valued parameter. The interval valued supply and demand parameters are also converted to real numbers by using parameters that do not necessarily depend on time. This procedure is depicted in the first subsection of this section. After that, an algorithm is presented to solve the proposed MOTP in later subsection.

### 5.1. Reduction of Interval into Real Number

In the proposed Model 4, the transportation cost is of the interval type, i.e., $C_{ij}^{\prime k} = [C_{ij}^{kl}, C_{ij}^{ku}]$; this means that there are some reasons for which the cost may take any value in the prescribed interval. Let us consider a parameter which depends on time. Let $t_0$ be the time assigned by the DM as the minimum range period. If the delivery occurred within the minimum range period, then the minimum transporting cost $C_{ij}^{kl}$ has to be paid. Due to delay of delivery the product, the cost becomes $\overline{C}_{ij}^k = C_{ij}^{kl}(1 - \lambda^{C_{ij}^k}) + C_{ij}^{ku}\lambda^{C_{ij}^k}$. Here, $\lambda^{C_{ij}^k}$ is a parameter for each $k$ such that

$$\lambda^{C_{ij}^k} = \begin{cases} 0, & \text{if } T_{ij} < t_0 \\ \frac{T_{ij} - t_0}{T_{ij}}, & \text{if } T_{ij} \geq t_0 \end{cases} \tag{5}$$

where $T_{ij}$ is the time of transportation from $i$th origin to $j$th destination and, then, $\lambda^{C_{ij}^k}$ is an increasing function of time.

Furthermore, all the objective functions in the considered MOTP are not the type to deliver goods on a scheduled time. It may also be about the profit by delivering the goods. Basically, when goods are required on an urgent basis, overtime duties of the entire working staff in the production system are required and, then, production cost becomes larger; consequently, DM expects a larger profit as requirement is fulfilled in hurry. In that case, if the goods are required in minimum time $t_0$, then the DM attains the maximum profit. Here, it is assumed that, if duration of ordering and purchasing time are $t_0$, then profit is $C_{ij}^{ku}$, which is the maximum value; otherwise, the value of production cost becomes $\overline{C}_{ij}^k = C_{ij}^{kl}\lambda^{C_{ij}^k} + C_{ij}^{ku}(1 - \lambda^{C_{ij}^k})$.

Pollution factor of carbon emission is incorporated into one of the objective functions in our formulated MOTP. It is observed that pollution factor increases when using more power required on an emergency basis. As the pollution factor is not a crisp quantity, we choose here the interval valued pollution factor by the following way: $\overline{C}_{ij}^k = C_{ij}^{kl}\lambda^{C_{ij}^k} + C_{ij}^{ku}(1 - \lambda^{C_{ij}^k})$. This indicates that, if $T_{ij} = t_0$,

then the pollution factor takes maximum value and as much as $T_{ij}$ increases and, then, the pollution factor decreases.

**Proposition 1.** *The transportation cost $\overline{C}_{ij}^{k}$ (objective function with minimization type) attains minimum value when transportation time in the kth objective function for the ith origin to the jth destination tends to the minimum value and conversely.*

**Proof.** The interval cost $[C_{ij}^{kl}, C_{ij}^{ku}]$ in the *k*th objective function for the *i*th origin to the *j*th destination is reduced to a real valued cost $\overline{C}_{ij}^{k}$ by the following way:

Let $\overline{C}_{ij}^{k} = C_{ij}^{kl}(1 - \lambda^{C_{ij}^{k}}) + C_{ij}^{ku}\lambda^{C_{ij}^{k}}$. The value of $\overline{C}_{ij}^{k}$ tends to minimum value as $\lambda^{C_{ij}^{k}}$ tends to zero. Here, $\lambda^{C_{ij}^{k}}$ is a function of time $T_{ij}$ as stated in Equation (5). From Equation (5), it is clear that $\lambda^{C_{ij}^{k}}$ tends to the minimum value as the transportation time $T_{ij}$ tends to the fixed time $t_0$, which is assigned by DM. Therefore, $\overline{C}_{ij}^{k}$ tends to its minimum value as the transportation time $T_{ij}$ tends to $t_0$, i.e., the cost component $\overline{C}_{ij}^{k}$ attains minimum value when transportation time in the *k*th objective function for the *i*th origin to the *j*th destination tends to the minimum value.

Conversely, when the transportation time in the *k*th route for the *i*th origin to the *j*th destination tends to the minimum value, then Equation (5) suggests that $\lambda^{C_{ij}^{k}}$ tends to zero and then $\overline{C}_{ij}^{k}$ tends to the minimum value $C_{ij}^{kl}$. This evidences the proof of the proposition. □

DM can also choose the function according to this choice, but it should be time dependent as per our consideration in this paper. Again, the supply $\overline{a}_i (= [a_i^l, a_i^u])$ and demand $\overline{b}_j (= [b_j^l, b_j^u])$ parameters are also considered as interval numbers. The interval numbers reduce into real numbers by the following way: $\overline{a}_i = a_i^l(1 - \lambda^{a_i}) + a_i^u\lambda^{a_i}$ and $\overline{b}_j = b_j^l(1 - \lambda^{b_j}) + b_j^u\lambda^{b_j}$. Here, $\lambda^{a_i}$ and $\lambda^{b_j}$ are the parameters not necessarily related to time but may be linear or stochastic or fuzzy depending upon the choice of DM.

**Proposition 2.** *The profit $\overline{C}_{ij}^{k}$ (objective function with maximization type) achieves maximum value when profit in the kth objective function for the ith origin to the jth destination reaches to the maximum value and conversely.*

**Proof.** The proof is left to the reader. □

*5.2. Algorithm for Solving Time Variant MOTP*

To find a compromised solution of the model (i.e., **Model 4**) we proceed through the following steps.

**Step 1**　At first, we change the interval valued transportation parameters of objective function $z^k$ of MOTP from **Model 4** to simple real parameters of MOTP using the procedure described in Section 5.1.

**Step 2**　Formulate MOTP as follows:
　　　　**Model 5**

$$\text{optimize} \quad z^k = \sum_{i=1}^{m}\sum_{j=1}^{n} \overline{C}_{ij}^{k} x_{ij}$$

$$\text{subject to} \quad \sum_{j=1}^{n} x_{ij} \leq \overline{a}_i, \quad i = 1, 2, \cdots, m$$

$$\sum_{i=1}^{m} x_{ij} \geq \overline{b}_j, \quad j = 1, 2, \cdots, n$$

$$x_{ij} \geq 0, \quad \forall \ i \text{ and } j.$$

Here, $\overline{C}_{ij}^k$, $\overline{a}_i$, and $\overline{b}_j$ are reduced into real parameters.

**Step 3** Consider goals and respective deviations corresponding to each of the objective functions in Model 5. Convert Model 5 into the single-objective optimization problem by model GP.

**Step 4** Solve the constructed model GP, report the optimum compromise solution which is denoted as $X^*$, and report the optimal value of each of the objective functions.

**Step 5** The optimal allocation is $X^*$, so the minimum transportation time $T$ (cf., Roy and Maity [10]) is calculated by the following way:

$$\text{minimize } T = \sum_{i=1}^{m} \sum_{j=1}^{n} T_{ij} \chi_{ij}, \quad \text{where} \quad \chi_{ij} = \begin{cases} 0, & \text{if } x_{ij} = 0 \ \ in \ \ X^* \\ 1, & \text{if } x_{ij} \neq 0 \ \ in \ \ X^* \end{cases}$$

To solve MOITP using the presented algorithm, we use the following flowchart (see Figure 2). The flowchart also contains the required pseudo code to solve the time variant MOTP.

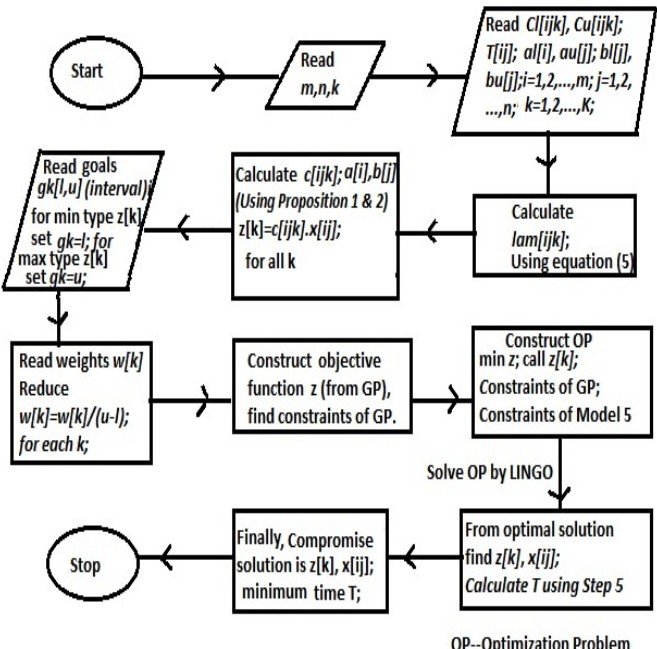

**Figure 2.** Flowchart to solve multi-objective interval transportation problem (MOITP).

**Proposition 3:** *The optimal solution of the model (i.e., Model 5) indicates a compromise solution of the objective functions $z^k$ and $T$ of* **Model 4** *.*

**Proof.** The proof is left to the interested reader. □

## 6. Case Study

In order to show the applicability of this paper, let us include the following case study.

A merchant has three rice meals for producing rice at three different locations, namely $A_1, A_2$, and $A_3$. Produced rice is sold to three dealers located in the cities $M_1, M_2$, and $M_3$. The problem is considered with three conflicting objective functions, namely transportation cost, profit, and pollution factor along with time corresponding to each of them. Here, transportation cost, profit, and pollution factor all are considered as interval values. Again, there is a time associated with each of the interval

parameters. Generally, time is an important factor in business. Here, time corresponding to each of the transportation cost intervals indicates that, if transportation is made in schedule time, then minimum value of interval transportation cost is considered; otherwise, transportation cost increases. Again profit relates to the fact that, if the requirement of goods is fulfilled in schedule time, then profit is the maximum value in terms of interval. As time becomes larger, the corresponding profit becomes smaller. Again, if the production and transportation are made in a hurry, then the pollution factor takes the maximum value of the interval valued pollution factor. In the aforementioned way, time is connected with the transportation parameters. Also, DM can choose the time and parameters in relation to their preference. In this example, we consider the same time corresponding to each parameter in a node of the different objective functions; however, someone can take different times related to each objective function. Then calculation in total time will be changed accordingly.

Table 1 presents transportation costs (in $) in each route that are interval valued (these are considered due to increasing the fuel price, road tax etc.). Table 2 represents profits (in $) in each domain, and Table 3 considers pollution factor. Again, the production in factories and demands in the markets are not uniform and crisp due to availability of raw materials, weather conditions, market demands, etc., so supply and demand are treated as interval values. The interval valued supply in the stores $A_i$ ($i = 1, 2, 3$) are $a_1 = [2000, 2100]$, $a_2 = [1900, 2000]$, and $a_3 = [2100, 2200]$ and, similarly, for the demand parameters to the markets $M_j$ ($j = 1, 2, 3$) are $b_1 = [1900, 2100]$, $b_2 = [1900, 2000]$, and $b_3 = [2000, 2200]$, respectively. The merchant expects schedule time for transportation $t_0 = 5$ days. According to the experience in the business area, the merchant chooses the goals [53,000, 58,825] for transportation cost ($z^1$), [47,000, 51,800] for profit ($z^2$), and [14,275, 18,825] for pollution factor ($z^3$). In most of the situations, interval goal corresponding to objective function is chosen as its minimum and maximum possible values with respect to the same set of constraints. Here, the merchant wants to minimize total transportation cost, to maximize profit, and to minimize total pollution factor with a good care to minimize the total time.

**Table 1.** Transportation cost $C_{ij}^1$ (in $) per kg.

|     | M1       | M2       | M3       |
| --- | -------- | -------- | -------- |
| A1  | [5, 10]  | [6, 8]   | [7, 10]  |
| A2  | [8, 12]  | [10, 15] | [9, 14]  |
| A3  | [12, 15] | [10, 14] | [12, 18] |

**Table 2.** Profit $C_{ij}^2$ (in $) per kg.

|     | M1       | M2      | M3      |
| --- | -------- | ------- | ------- |
| A1  | [4, 6]   | [5, 7]  | [3, 6]  |
| A2  | [5, 7]   | [6, 8]  | [4, 7]  |
| A3  | [10, 12] | [8, 12] | [8, 10] |

The mathematical model is formulated corresponding to Tables 1–4 as follows:

**Model 6**

minimize $\quad z^1 = [5,10]x_{11} + [6,8]x_{12} + [7,10]x_{13} + [8,12]x_{21} + [10,15]x_{22} + [9,14]x_{23}$
$\qquad\qquad + [12,15]x_{31} + [10,14]x_{32} + [12,18]x_{33}$,

maximize $\quad z^2 = [4,6]x_{11} + [5,7]x_{12} + [3,6]x_{13} + [5,7]x_{21} + [6,8]x_{22} + [4,7]x_{23}$
$\qquad\qquad + [10,12]x_{31} + [8,12]x_{32} + [8,10]x_{33}$,

minimize $\quad z^3 = [2,4]x_{11} + [1,5]x_{12} + [3,4]x_{13} + [3,5]x_{21} + [2,4]x_{22} + [4,6]x_{23}$
$\qquad\qquad + [2,4]x_{31} + [1,4]x_{32} + [3,6]x_{33}$,

subject to $\quad \displaystyle\sum_{j=1}^{3} x_{1j} \leq [2000,2100]$,

$$\sum_{j=1}^{3} x_{2j} \leq [1900,2000],$$

$$\sum_{j=1}^{3} x_{3j} \leq [2100,2200],$$

$$\sum_{i=1}^{3} x_{i1} \geq [1900,2100],$$

$$\sum_{i=1}^{3} x_{i2} \geq [1900,2000],$$

$$\sum_{i=1}^{3} x_{i3} \geq [2000,2200],$$

$$x_{ij} \geq 0, \quad i = 1,2,3 \text{ and } j = 1,2,3.$$

Model 6 is the formulated MOITP corresponding to the numerical example.

**Table 3.** Pollution factor $C_{ij}^3$ per kg of goods.

|  | **M1** | **M2** | **M3** |
|---|---|---|---|
| A1 | [2, 4] | [1, 5] | [3, 4] |
| A2 | [3, 5] | [2, 4] | [4, 6] |
| A3 | [2, 4] | [1, 4] | [3, 6] |

**Table 4.** $T_{ij}$ (in day) associated with transportation parameters.

|  | **M1** | **M2** | **M3** |
|---|---|---|---|
| A1 | 15 | 10 | 8 |
| A2 | 6 | 4 | 5 |
| A3 | 4 | 6 | 3 |

*6.1. Discussion on Results*

We solve Model 6 with the help of the presented algorithm; the solutions are achieved for different weights, and they are listed in Table 5.

**Table 5.** Optimal solution of Model 6 for different weights.

| Weights | Optimal Solution | Optimal Values |
|---------|------------------|----------------|
| $w_1 = \frac{1}{3}, w_2 = \frac{1}{3}, w_3 = \frac{1}{3}$ | $x_{12} = 1900, x_{23} = 1800, x_{31} = 1900, x_{33} = 200,$ and other variables are zero. | $Z^1 = 54{,}700, Z^2 = 48{,}800,$ $Z^3 = 16{,}350, T = 22.$ |
| $w_1 = \frac{1}{4}, w_2 = \frac{1}{2}, w_3 = \frac{1}{4}$ | $x_{12} = 2100, x_{23} = 2000, x_{31} = 2100$ and other variables are zero. | $Z^1 = 57{,}900, Z^2 = 51{,}800,$ $Z^3 = 17{,}450, T = 19.$ |
| $w_1 = \frac{1}{4}, w_2 = \frac{1}{4}, w_3 = \frac{1}{2}$ | $x_{12} = 1900, x_{23} = 1800, x_{31} = 1900, x_{33} = 200,$ and other variables are zero. | $Z^1 = 54{,}700, Z^2 = 48{,}800,$ $Z^3 = 16{,}350, T = 22.$ |
| $w_1 = \frac{1}{8}, w_2 = \frac{1}{8}, w_3 = \frac{3}{4}$ | $x_{13} = 1800, x_{22} = 1900, x_{31} = 1900, x_{33} = 200,$ and other variables are zero. | $Z^1 = 58{,}825, Z^2 = 48{,}775,$ $Z^3 = 14{,}275, T = 19.$ |
| $w_1 = \frac{1}{10}, w_2 = \frac{1}{10}, w_3 = \frac{4}{5}$ | $x_{13} = 1800, x_{22} = 1900, x_{31} = 1900, x_{33} = 200,$ and other variables are zero. | $Z^1 = 58{,}825, Z^2 = 48{,}775,$ $Z^3 = 14{,}275, T = 19.$ |
| $w_1 = 0, w_2 = 0, w_3 = 1$ | $x_{13} = 1800, x_{22} = 1900, x_{31} = 1900, x_{33} = 200,$ and other variables are zero. | $Z^1 = 58{,}825, Z^2 = 48{,}775,$ $Z^3 = 14{,}275, T = 19.$ |
| $w_1 = 0, w_2 = 1, w_3 = 0$ | $x_{12} = 2100, x_{23} = 2000, x_{31} = 2100,$ and other variables are zero. | $Z^1 = 57{,}900, Z^2 = 51{,}800,$ $Z^3 = 17{,}450, T = 19.$ |
| $w_1 = 1, w_2 = 0, w_3 = 0$ | $x_{11} = 275, x_{12} = 1825, x_{21} = 1625, x_{22} = 75,$ $x_{33} = 1700$ and other variables are zero. | $Z^1 = 53{,}000, Z^2 = 42{,}767,$ $Z^3 = 20{,}596, T = 28.$ |

From Table 5, we see that the obtained solutions are satisfactory based on the relative importance of the objective functions, i.e., the priority of the weights. Although, we list fewer, one can set weights for their preferred objective functions to get better optimal solutions. We see that, when weights are the same (i.e., $w_i = \frac{1}{3}$, $i = 1$, 2, 3), then satisfactory levels of optimal solutions for $z^1$, $z^2$, and $z^3$ are 80%, 40%, and 70% respectively. The total time of transportation is 22 days. The time is calculated through the procedure given in the algorithm, and we do not consider any objective function corresponding to time. Meanwhile, if the minimization of total time of transportation is considered, then the solution corresponding to weights 0.25, 0.5, and 0.25 is the best one, although in that solution, the achievement of goals becomes lesser in compare to the solutions of equal weights. If the merchant wishes to minimize time as the first priority, then the minimum value of T becomes 16 days. In that case, the transportation cost and pollution factor both increase and profit decreases significantly. In the proposed model, the optimal solution minimizes total time with satisfactory levels of goals. Another important factor is that the sustainable development of the atmosphere is made through minimizing the pollution factor. If the sustainable development is given the best priority, then weight of $z^3$ increases and then transportation cost becomes larger with profit decreasing, which is also presented in Table 5. Therefore, we are able to say that this study minimizes time as well as finds a better issue for sustainable development of the atmosphere in the optimization era.

*6.2. Sensitivity Analysis*

The study endeavours the time variant MOTP. The most important factors in this study are how the time is correlated with the other objective functions and finding the optimal compromise solution. Basically, time in a MOTP makes a difference in profit and cost of transportation. Here, we provide how the time plays an important role in the MOTP.

The obtained solution (cf., first solution in Table 5) shows that allocation is made in the cell $(1, 3)$ in which transportation cost is $[7, 10]$ and the transportation time is 8 days. There are low transportation costs $[5, 10]$ and $[6, 8]$ in the cells $(1, 1)$ and $(1, 2)$, respectively. If DM wishes to minimize transportation cost, only then they must choose the minimum among the costs. In our presented study, time has reduced the transportation parameters in such a way that the reduced values act unbiased for both the seller and buyer sides. In this paper, the proposed model removes the complexities for optimizing the objective functions along with minimizing the time.

Our proposed methodology is not comparable directly with any other existing techniques for solving MOITP like fuzzy programming, weighting method, utility function approach, etc. One can wish to compare our technique with existing methodologies for solving MOTP like fuzzy programming, weighting method, utility function, etc. He/she will have to consider some assumptions to formulate the mathematical model for given MOITP.

In case of minimization of transportation cost without considering profit and pollution factor, DM sets the weights 1, 0, and 0 corresponding to objective functions $z^1$, $z^2$, and $z^3$, respectively. In that case, the minimum transportation cost is $\$53,000$. Then, the profit decreases significantly along with the larger value of pollution factor. Furthermore, when DM wants to find maximum profit without considering the other objective functions, then the weights are taken as 0, 1, and 0 corresponding to objective functions $z^1$, $z^2$, and $z^3$, respectively. In that case, the maximum profit is $\$51,800$. This is the maximum possible profit achieved by DM. Finally, to extract the solution of better pollution factor, DM selects the weights 0, 0, and 1 corresponding to objective functions $z^1$, $z^2$, and $z^3$, respectively. In that case, the objective value of pollution factor is $\$14,275$. This is the case of the solution in favour of minimizing the pollution factor. Considering the above discussion, it is clear that the optimum solutions of objective functions depend on the preferences of the objective functions. Therefore, sustainable development through transportation is possible by minimizing pollution factor in almost every real-life transportation system. In the presented problem, the solution regarding equal weights is a better compromise solution.

Now, to justify that the obtained solution of our proposed methodology is a better solution, let us introduce a utility function in the form

$$f(g) = w_1 \frac{\bar{z}^1 - z^1}{\bar{z}^1 - \underline{z}^1} + w_2 \frac{\bar{z}^2 - z^2}{\bar{z}^2 - \underline{z}^2} + w_3 \frac{\bar{z}^3 - z^3}{\bar{z}^3 - \underline{z}^3} + w_4 \frac{\overline{T} - T}{\overline{T} - \underline{T}},$$

where $\bar{z}^k$ = maximum value of $z^k$ for $k$th ($k = 1, 2, 3$) objective function, $\overline{T}$ = maximum possible value of $T$, $\underline{z}^k$ = minimum value of $z^k$ for $k$th objective function, $\underline{T}$ = minimum transportation time, and $w_k$ and $w_4$ are weights for the objective function $z^k$ and time T. The value of the utility function $f$ lies between 0 and 1. The bigger value of $f$ proposes a better compromise solution of the MOTP. Considering the equal weights for the objective functions, we get $f(g) = 0.60$. This value suggests that the obtained solution is a better compromise solution.

## 7. Concluding Remarks and Outlook

This work has presented an efficient and scalable model for solving time variant multi-objective transportation problems. The considered problem has been modelled as a MOTP, which can be efficiently solved considering time minimization using the goal programming approach. The framework and solution concept have been extended to capture sustainable development using interval valued transportation parameters.

The focus of this paper has been on developing a mathematical model with efficient precise solutions. A new algorithm has been included for solving time variant MOTP using goal programming. The solution of case study has been analysed under different situations. Until today, researchers have used the methodology for solving MOTP to optimize the objective function by considering time as one of the objective functions, but here, we have determined an optimal (compromise) solution of MOTP that minimizes the total time without considering objective function of time in the model of MOTP. Another important factor of this study has been considered sustainable development by minimizing the pollution factor. Realizing global warming and overall pollution, it has been utmost essential to keep our nature neat and clean for us to not look at maximum profit, which creates harmful effects on the atmosphere.

The main limitations of the study are as follows: One of the most important drawbacks in GP is the selection of the goals. There may be a situation in which, if the goal is not selected in proper way,

then the solution is infeasible. Another important factor is to reduce the value of objective functions in same scale; otherwise, the solution will be the worst optimal (compromise) solution.

In future scope of study, MOITP should be integrated in different areas of study such as optimal network designing, rail and passenger transportation, and fare payment systems, etc. Again, the proposed model of MOITP can be used in selection of modes in a variety of transportation improvement policies such as mobility management strategies, pricing reforms and smart growth land use policies, minimizing carbon emission in industrial systems, etc. In addition to the above, the proposed study can be implemented in different uncertain environments to accommodate real-life situations for selecting optimal decisions considering the sustainable development of atmosphere.

**Author Contributions:** Conceptualization, G.M., S.K.R. and J.L.V.; Data curation, S.K.R.; Sensitivity analysis, J.L.V.; Methodology, G.M.; Writing-original draft, G.M.; Writing-review and editing, S.K.R. and J.L.V.

**Funding:** The research of Jose Luis Verdegay is supported in part by the project, financed with FEDER funds, TIN2017-86647-P from the Spanish Govern.

**Acknowledgments:** The Research of Jos*é* Luis Verdegay is supported in part by project TIN2017-86647-P (Spanish Ministry of Economy and Competitiveness, includes FEDER funds from the European Union). The authors are very much thankful to the anonymous reviewers for providing the precious comments that helped so much too rigorously increase the quality of the paper.

**Conflicts of Interest:** The authors declare no conflict of interest.

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
