# Peer review of "Time Variant Multi-Objective Interval-Valued Transportation Problem in Sustainable Development"

_sustainability, doi:10.3390/su11216161_

Round 1
Reviewer 1 Report
The authors apply goal programming to solve the Multi-Objective Transportation Problem, which also considers the pollution factor as an objective to be minimized. The paper is quite inserting but there are some grammatical errors that make the paper sometimes hard to read and understand, therefore it should be proofread. Below are my comments:
The paper should be proofread. The paper does not make a clear case of why goal programming applied instead of multi-objective optimization. In section 3. it is mentioned that production cost is considered as an objective function but is not considered in the case study. “Optimal solution of a MOTP is basically referred to as Pareto optimal solution. Here, we find the Pareto optimal solution using goal programming “ Multi-objective problems don’t have a single optimal solution but a set called the Pareto front. You limited your search to a single Pareto optimal solution by applying goal programming. This, however, doesn’t assure that the Pareto optimal solution will always be obtained. Since the objectives have a different scale, have they been normalized before weights were applied? Section 5.2 describes the algorithm used for solving the MOTP. The section should display the algorithm’s pseudocode to better understand how it works. It is only mentioned that goal programming is used to obtain the optimum solution. But how does goal programming work? Are there different approaches? If so, which one was chosen and why? In section 6. cities D1, D2, and D3 are mentioned but never used in tables or equations. It would be interesting to see the time required to obtain an optimal solution.
Author Response
Response Sheet about Reviewer’s Comments
Journal Name: Sustainability
Manuscript ID.: Sustainability-602488
Paper Title: Time variant multi-objective interval-valued transportation problem in
sustainable development.
Authors: Gurupada Maity, Sankar Kumar Roy and Jose Luis Verdegay
Respected Reviewer,
The revised manuscript is thoroughly checked with great care and rewritten on the basis of the reviewer’s comments in the paper “Time variant multi-objective interval-valued transportation problem in sustainable development”. In the following, please find the responses regarding the reviewer’s comments point-wise and these are reflected in blue-color in the revised manuscript.
Reviewer’s Comments
# Reviewer 1:
Comments and Suggestions for Authors
The authors apply goal programming to solve the Multi-Objective Transportation Problem, which also considers the pollution factor as an objective to be minimized. The paper is quite interesting but there are some grammatical errors that make the paper sometimes hard to read and understand, therefore it should be proofread. Below are my comments:
Query: The paper should be proofread.
Response: We have checked the revised manuscript thoroughly and made many changes to read and understand the paper comfortably to the readers.
Query: The paper does not make a clear case of why goal programming applied instead of multi-objective optimization.
Response: We have discussed in detail why goal programming is used to solve our presented problem (please see page 3) in the revised manuscript.
Query: In Section 3. it is mentioned that production cost is considered as an objective function but is not considered in the case study. “Optimal solution of a MOTP is basically referred to as Pareto optimal solution. Here, we find the Pareto optimal solution using goal programming “ Multi-objective problems don’t have a single optimal solution but a set called the Pareto front. You limited your search to a single Pareto optimal solution by applying goal programming. This, however, doesn’t assure that the Pareto optimal solution will always be obtained.
Response: We have omitted the objective function corresponding to production cost as it is not considered in case study. But it is a general MOTP, so there is no problem to consider production cost as an objective function. We have changed the Pareto optimal solution into compromise solution. The sentences have been rewritten and we have presented why goal programming is appropriate to find compromise solution in the revised manuscript (see page 3).
Query: Since the objectives have a different scale, have they been normalized before weights were applied?
Response: We have discussed about the normalization of different scales in the revised manuscript (please see page 4). The authors are very much thankful to the anonymous reviewer for pointing this mistake.
Query: Section 5.2 describes the algorithm used for solving the MOTP. The section should display the algorithm’s pseudo code to better understand how it works.
Response: We have included a graphical representation (see flowchart in Figure 2) which includes pseudo code for solving the considered problems using the presented algorithm.
Query: It is only mentioned that goal programming is used to obtain the optimum solution. But how does goal programming work? Are there different approaches? If so, which one was chosen and why?
Response: We have utilized Goal programming to solve multi-objective optimization problem. How goal programming works to solve MOTP, and why the technique is appropriate has been discussed in the Introduction in the revised manuscript (please see pages 3 and 4).
Query: In section 6. cities D1, D2, and D3 are mentioned but never used in tables or equations. It would be interesting to see the time required to obtain an optimal solution.
Response: We have changed the cities D1, D2, and D3 to cities M1, M2, and M3 and they are used in the tables of Section 6 in the revised manuscript. Time is included in the problem and plays an important role which is discussed in the revised manuscript (please see pages 6 and 11). Authors are very much thankful to the anonymous reviewer for providing the comments.
Finally, the authors are very much thankful to the anonymous reviewer for his precious comments that helped us too much to rigorously improve the quality of the paper.
Reviewer 2 Report
This study proposes a model for time variant multi-objective interval-valued transportation problem. The study discusses the relevant information and literature. The proposed model is described afterwards. The numerical experiments are performed next. Some managerial insights and supporting discussions are given as well. In general, I think that the paper fits well into the scope of the journal. However, some revisions are required before the paper can be considered for publication. Certain segments of the paper must be strengthened. Below please find more specific comments:
*Page 1 line 4: “taking care about to sustain the environment for better living in future” does not sound well. I suggest using “considering negative externalities for the environment and quality of life for future generations”.
*Page 1 line 11: “A case study is considered” does not sound well. I suggest using “A case study is conducted”.
*Page 1: The authors start the introduction section with a short discussion, highlighting the importance of transportation. However, this discussion should be strengthened. In particular, the authors should clearly highlight the importance of both passenger and freight transportation modes and their critical role for the economic development of numerical countries. Also, the discussion should be strengthened by adding the following references, which highlight importance of passenger and freight transportation.
Liu, L. and Chen, R.C., 2017. A novel passenger flow prediction model using deep learning methods. Transportation Research Part C: Emerging Technologies, 84, pp.74-91.
Wagenaar, J., Kroon, L. and Fragkos, I., 2017. Rolling stock rescheduling in passenger railway transportation using dead-heading trips and adjusted passenger demand. Transportation Research Part B: Methodological, 101, pp. 140-161.
Dulebenets, M.A., 2018. The vessel scheduling problem in a liner shipping route with heterogeneous fleet. International Journal of Civil Engineering, 16(1), pp.19-32.
Jalalian, M., Gholami, S. and Ramezanian, R., 2019. Analyzing the trade-off between CO2 emissions and passenger service level in the airline industry: Mathematical modeling and constructive heuristic. Journal of Cleaner Production, 206, pp. 251-266.
Dulebenets, M.A., 2018. Green vessel scheduling in liner shipping: Modeling carbon dioxide emission costs in sea and at ports of call. International Journal of Transportation Science and Technology, 7(1), pp.26-44.
Zhen, L., Liang, Z., Zhuge, D., Lee, L. H., & Chew, E. P. (2017). Daily berth planning in a tidal port with channel flow control. Transportation Research Part B: Methodological, 106, 193-217.
Xiang, X., Liu, C., & Miao, L. (2018). Reactive strategy for discrete berth allocation and quay crane assignment problems under uncertainty. Computers & Industrial Engineering, 126, 196-216.
Page 3: At the end of the introduction section please add a paragraph, defining the structure of the manuscript (i.e., what the readers should expect in the following sections).
*Page 4: An illustrative figure somewhere in section 3 will help the readers to visualize the problem.
*Page 4: In section 4 of the papers, the authors show some representative formulations for the transportation problem. However, no references are provided. Please support selection of these particular models with adequate references.
*Page 8: The authors should elaborate more on the input data selection for testing the proposed methodology. Do you adopt the data from the literature? Or certain values have been collected from other sources? If the literature is used, the appropriate references should be provided.
*Page 11: The conclusion section should be strengthened. The authors should clearly highlight limitations of this study and how they will be addressed in future research.
Author Response
Response Sheet about Reviewer’s Comments
Journal Name: Sustainability
Manuscript ID.: Sustainability-602488
Paper Title: Time variant multi-objective interval-valued transportation problem in
sustainable development.
Authors: Gurupada Maity, Sankar Kumar Roy and Jose Luis Verdegay
Respected Reviewer,
The revised manuscript is thoroughly checked with great care and rewritten on the basis of the reviewer’s comments in the paper “Time variant multi-objective interval-valued transportation problem in sustainable development”. In the following, please find the responses regarding the reviewer’s comments point-wise and these are reflected in blue-color in the revised manuscript.
Reviewer’s Comments
Reviewer #2:
Comments and Suggestions for Authors
This study proposes a model for time variant multi-objective interval-valued transportation problem. The study discusses the relevant information and literature. The proposed model is described afterwards. The numerical experiments are performed next. Some managerial insights and supporting discussions are given as well. In general, I think that the paper fits well into the scope of the journal. However, some revisions are required before the paper can be considered for publication. Certain segments of the paper must be strengthened. Below please find more specific comments:
Query: *Page 1 line 4: “taking care about to sustain the environment for better living in future” does not sound well. I suggest using “considering negative externalities for the environment and quality of life for future generations”.
Response: We have changed the sentence according to the respected reviewer’s suggestion in the revised manuscript. The authors are very much thankful to the anonymous reviewer for his positive thinking.
Query: *Page 1 line 11: “A case study is considered” does not sound well. I suggest using “A case study is conducted”.
Response: We have rewritten the mentioned sentence according to the suggestion of reviewer in the revised manuscript. The authors are grateful to the anonymous reviewer for his carefully reading the manuscript.
Query: *Page 1: The authors start the introduction section with a short discussion, highlighting the importance of transportation. However, this discussion should be strengthened. In particular, the authors should clearly highlight the importance of both passenger and freight transportation modes and their critical role for the economic development of numerical countries. Also, the discussion should be strengthened by adding the following references, which highlight importance of passenger and freight transportation.
Liu, L. and Chen, R.C., 2017. A novel passenger flow prediction model using deep learning methods. Transportation Research Part C: Emerging Technologies, 84, pp.74-91.
Wagenaar, J., Kroon, L. and Fragkos, I., 2017. Rolling stock rescheduling in passenger railway transportation using dead-heading trips and adjusted passenger demand. Transportation Research Part B: Methodological, 101, pp. 140-161.
Dulebenets, M.A., 2018. The vessel scheduling problem in a liner shipping route with heterogeneous fleet. International Journal of Civil Engineering, 16(1), pp.19-32.
Jalalian, M., Gholami, S. and Ramezanian, R., 2019. Analyzing the trade-off between CO2 emissions and passenger service level in the airline industry: Mathematical modeling and constructive heuristic. Journal of Cleaner Production, 206, pp. 251-266.
Dulebenets, M.A., 2018. Green vessel scheduling in liner shipping: Modeling carbon dioxide emission costs in sea and at ports of call. International Journal of Transportation Science and Technology, 7(1), pp.26-44.
Zhen, L., Liang, Z., Zhuge, D., Lee, L. H., & Chew, E. P. (2017). Daily berth planning in a tidal port with channel flow control. Transportation Research Part B: Methodological, 106, 193-217.
Xiang, X., Liu, C., & Miao, L. (2018). Reactive strategy for discrete berth allocation and quay crane assignment problems under uncertainty. Computers & Industrial Engineering, 126, 196-216.
Response: We have added some useful sentences to the Introduction for increasing the novelty by mentioning several types of transportation situations and utility of multi-objective optimization problem. The literature review section has been enriched using the following recent researches (please see references [5]-[8], [19], [21] and [22]).
Query: Page 3: At the end of the introduction section please add a paragraph, defining the structure of the manuscript (i.e., what the readers should expect in the following sections).
Response: In the revised manuscript, we have added a paragraph which informs to the readers about the contents of the rest of the research (see page 3).
Query: *Page 4: An illustrative figure somewhere in section 3 will help the readers to visualize the problem.
Response: We have presented a figure (please see Figure 1) in Section 3 to visualize the transportation problem.
Query: *Page 4: In section 4 of the papers, the authors show some representative formulations for the transportation problem. However, no references are provided. Please support selection of these particular models with adequate references.
Response: We have presented the mathematical model of TP in Section 4. We have added the references ([2], [10]) to the respective position in the revised manuscript.
Query: *Page 8: The authors should elaborate more on the input data selection for testing the proposed methodology. Do you adopt the data from the literature? Or certain values have been collected from other sources? If the literature is used, the appropriate references should be provided.
Response: The data is given hypothetically based on the real-life scenario to test the efficiency of the present study.
Query: *Page 11: The conclusion section should be strengthened. The authors should clearly highlight limitations of this study and how they will be addressed in future research.
Response: The conclusion section is rewritten to indicate the limitations of the proposed study. Also, we have addressed future research directions to the conclusion section in revised manuscript.
Finally, the authors are very much thankful to the anonymous reviewers for their precious comments that helped us too much to rigorously improve the quality of the paper.

Round 2
Reviewer 2 Report
The authors have adequately addressed my original comments. Presentation of the manuscript has been improved. Therefore, I recommend acceptance.